# Young Consumers' Intention to Participate in the Sharing Economy: An Integrated Model

José Alberto Martínez-González *, Eduardo Parra-López and Almudena Barrientos-Báez

Department of Business Management and Economic History, Faculty of Economics, Business and Tourism, University of La Laguna, 38200 San Cristóbal de La Laguna, Spain; eparra@ull.edu.es (E.P.-L.); almudenabarrientos@iriarteuniversidad.es (A.B.-B.)
* Correspondence: jmartine@ull.edu.es

**Abstract:** This paper aims to analyze the external and internal drivers of young consumers' intention to participate in the sharing economy in tourism. From previous findings, a causal model (PLS) is designed to generate an integrated, practical, and novel structural model that significantly predicts the intention to participate. The model, consisting of nine dimensions, includes consumers' external and internal variables. Separately, these variables have all been considered relevant in the literature, though they have not been studied jointly before. The descriptive results show the excellent attitude and predisposition of young people toward the tourism sharing economy, which facilitates their participation. Through the model, the importance of all internal and external consumer variables in the formation of intention are proven; however, attitude and social norm are most notable among them. Trust is also a critical variable that serves as the link between internal and external variables. The study provides managers of sharing economy platforms with knowledge to encourage young consumers' participation in a communication and market orientation context. The generational approach (Generation Z) used also allows the conclusions and implications to be transferred to other regions and sectors.

**Keywords:** sharing economy; consumer variables; consumer satisfaction; intention to participate; sharing values; young consumers' attitude; Generation Z

## 1. Introduction

The sharing economy is considered to be a new business paradigm and a new form of commerce [1–3]. There is no universally accepted definition of the sharing economy, and different terms are used to refer to the same phenomenon [4,5]. In the complete and recent conceptual revision carried out by Schlagwein, Schoder, and Spindeldreher in 2020, the sharing economy is defined as a commercial or non-commercial peer-to-peer model facilitated by an intermediary [3] (p. 12). Such is the growth in this sharing model that it is predicted that by 2025, the global volume of operations will have reached 335 billion dollars [3]. The reasons for the rapid growth of this new sharing culture has been strongly related to consumers' changing attitudes, values, and behavior that have arisen following the recent economic crisis [4,6]. In particular, the tourism sector has been one of the first to develop the sharing economy and is the second most important in terms of global business volume [7–9]. The authenticity of social interaction with residents in the tourism destination has been a determining factor in the evolution of the sharing economy in tourism [10–12]. It should also be noted that the tourism industry possesses ideal characteristics for sharing economy platforms (e.g., Airbnb, Uber, BlaBlaCar) such as widely available resources (e.g., houses, rooms, beds) and affordable prices [13,14].

In all sectors, the sharing economy depends on consumers' intention to participate, as shown in literature reviews and definitions provided by authors and institutions of recognized prestige (e.g., European Commission, Oxford Dictionary) [15–17]. It was shown that the intention to participate is an essential element of the sharing economy and the

best predictor of consumers' behavior [2,18,19]. Despite its importance, its study has been relegated to the background, due to the attention given to global and structural economic aspects [8,15,19]. With few exceptions [20], previous studies on the sharing economy have focused mainly on supply and demand issues, traditional business, or society [21].

The interest in the intention to participate in the tourism sharing economy is even more significant regarding young consumers. This is the case in Generation Z, which includes individuals born in 1995 or later [22]. Like Millennials, Generation Z is 'driving' the growth of the sharing economy and is increasingly influencing social and economic conditions [23,24]. There is high agreement in the literature that young consumers share similar attitudes, perceptions, values, and behavior [25,26]. Although studies on the participation of Generation Z in the sharing economy are scarce, this population segment has ideal characteristics because of their technological and digital nature [27]. In addition, they are communicative, collaborative, worried about social and environmental problems and have an identity based less on possessions and more on relationships [28]. University students are particularly good representatives of young consumers, and they have the knowledge and skills to make decisions and are willing to adopt behavior related to the sharing economy [29–31]. Moreover, they are very representative of young digital consumers [32]. However, the generational approach must be taken with some caution. There is still an open debate about the global validity of the results of studies with younger generation and there are few studies carried out with Generation Z [33]. Finally, studies based on age and educational level are useful, as these variables do influence consumers' perceptions, intentions, and behavior [34–36].

To fill certain gaps in the literature, this study investigates young consumers' intention to participate in the sharing economy in tourism, enriching the theoretical and empirical literature on this field. It is important to note that no specific sharing economy site or platform is studied, but rather the intention to use and participate in the sharing economy in tourism through the most relevant sites are analyzed. In addition, we use young consumers' intention as the best predictor of behavior, owing to the difficulties of obtaining real data about their future support [37,38]. A complete causal model is developed that is novel, practical, and able to predict young consumers' intention to participate in the sharing economy. This model can be used by tourism companies and institutions to improve their online marketing orientation and actions. Moreover, the proposed model uses consumers' external and internal variables that have not previously been studied together. It includes variables and relationships that the literature on this field considers relevant, together with variables from other theories and causal models of reference. In addition, the generational approach can facilitate the adoption of online marketing actions at a global level. An Importance-Performance Analysis (IPMA) is included to improve the model.

## 2. Literature Review

It is still not clear why consumers participate in sharing economy platforms, or, conversely, why they might be reluctant to do so [2,19,39]. However, it was proven that intention is the best predictor of participation, and many variables determine its formation, given the complexity of consumers' decisions and behavior [40]. Regarding its definition, intention is a conscious decision on the part of a consumer to behave in a certain way after considering the available information [41]. Intention also refers to the effort made to carry out this behavior. Thus, the greater the intention, the greater the effort [42].

Several authors have classified previous studies on consumer intention and behavior in the sharing economy. According to Xu, Zeng, and He [43], these studies can be categorized into four groups, depending on their main topic:

(a) Consumers' motivation.
(b) Consumers' intention to participate in the sharing economy through traditional theories.
(c) Different influential factors in consumers' intention to participate in the sharing economy through lesser known theories.

(d) Models based on previous theories incorporating other variables and complementary relationships.

Within the first approach, it is considered that motivation is a crucial variable for consumers' participation in the sharing economy. It should be noted that motivational factors differ from those belonging to traditional commerce or for various socio-demographic groups [2,44,45]. Milanova and Maas [46] suggest that factors that motivate consumer participation in the sharing economy are social, economic, and technological. More recently, Niezgoda and Kowalska [4] support this view by highlighting personal, social, and ideological factors. Specifically, Tussyadiah [47] found that sustainability, community, and economic benefits are three main factors that motivate users to stay in Airbnb accommodations. On the other hand, the motivational factors that favor the intention to participate in the sharing economy have also been classified as internal (intrinsic) and external (extrinsic) to the consumer. The former is related to the satisfaction inherent in participation, and the latter is related to the results of such participation [8,15,48]. There is a clear predominance of extrinsic over intrinsic motivation, both in the case of suppliers and consumers, although it is not necessarily a question of monetary motivation [49,50].

The second approach to the study of consumers' intention to participate in the sharing economy in tourism includes four well-known related theories. First, there is the Theory of Reasoned Action (TRA) [51] in which attitudes and social norms determine intention. However, the omission in the TRA of certain non-volitional factors (e.g., resources) has questioned its applicability in the context of consumer behavior, and this has favored the emergence of new theories [42,52]. Second, the Theory of Planned Behavior (TPB) [37], which is based on TRA, has been validated in several studies. It is probably one of the most referenced theories to explain consumers' intentions and behavior in the sharing economy [53,54]. TPB assumes that subjective norms, attitudes toward the behavior and the control of perceived behavior predict the intention to participate [55,56]. Despite being a reference model, some authors have found this theory has low predictive power [40]. Third, the Technology Acceptance Model (TAM) [57] is an authoritative model for the adoption of information technology products [54,58]. TAM is based both on TRA and TPB, and it uses two constructs. The perceived utility is related to the degree in which a person believes that the use of a particular system may improve his/her performance. On the other hand, perceived facility is related to the degree in which a person believes that is easy to use an information system [57,59]. Although TAM could be useful in the online context of the sharing economy, it simplifies the influence of human relationships by not considering subjective norms [60]. Finally, and not strictly related to TRA, TPB or TAM, some authors have used the Social Exchange Theory (SET) [61] to predict the intention to participate in the sharing economy [62]. According to SET, consumers participate in the sharing economy because economic resources (e.g., product, service, and knowledge) and social resources (e.g., friendship and reputation) can be exchanged obtaining benefits reciprocally [63,64].

The third approach in the study of the intention to participate in the sharing economy in tourism includes a complete set of lesser known theories, different from the previous ones and rarely used [43]. For example, the Prospect Theory (PT) [65] descriptively addresses consumers' intentions under conditions of risk and uncertainty, focusing on the perceived value of gains and, especially, losses [66,67]. Alternatively, Kim, Wu, and Nam [42] studied the intention to participate in the sharing economy in tourism through the Normal Activation Model (NAM) [68]. NAM examines altruistic or prosocial intention and behavior and is one of the most prominent theories in the context of socially or environmentally responsible behavior [69,70]. NAM uses three primary constructs: awareness of the negative consequences that behavior has for other people; the feeling of responsibility for not acting prosocially; and the moral obligation to act or not in a prosocial environment [71–73]. By contrast, Zhu, So and Hudson [54] analyzed the drivers of consumer behavior in the sharing economy through the Self-Efficacy Theory (SEFT) [74,75]. According to SEFT, people are self-organized and proactive, with consumer behavior being

the result of the interaction between personal factors and the environment. In the model, self-efficacy plays a fundamental role, and it is defined as individual judgments about a person's abilities to carry out a behavior successfully. Finally, Akbar and Tracogna [76] analyzed consumer behavior in the sharing economy through the Transaction Cost Theory (TCT) [77]. TCT focuses on sets of contractual arrangements to manage economic transactions in the presence of transaction costs, bounded rationality, and opportunism. The key variables of TCT refer to the nature and characteristics of economic transactions: frequency, uncertainty, and asset specification.

The fourth approach that addresses consumers' intentions in the tourism sharing economy has emerged due to the digital nature of young consumers, the intangible nature of tourism, and other limitations and critical factors mentioned within the three approaches above [39]. Theories and models belonging to this approach try to reflect the complexity of consumers' behavior. They include new internal and external variables in an integrated manner, together with certain variables that belong to some of the mentioned models above (e.g., attitude, social norm). Moreover, these variables and the causal relationships have been verified in previous studies [18], though it is worth noting that studies in a tourism context employing this approach are extremely scarce [78]. Belonging to this fourth approach, Na and Kang [79] designed a predictive model of intention using resource characteristics, core competencies, shared values, and distinctive competitive advantage as independent constructs. The model by Yang, Lee, Lee, and Koo [80] included trust, reputation, and security, among other variables. Whereas Nadeem, Juntunen, Hajli, and Tajuidi [81] presented their causal model taking into account ethical perceptions and their influence on consumers' participation and intention to co-create. Gupta, Esmaeilzadeh, Uz, and Tennant [78] took culture into account, while Davlembayeva, Papagiannidis, and Alamanos [82] included social factors in their causal model, and Nadeem and Al-Imamy [83] focused on the role of ethics. As mentioned, and as shown by the above authors, this approach attempts to account more completely for the complexity of consumers' behavior. With this in mind, it was decided to adopt this fourth approach in this study by incorporating new variables and relationships along with aspects of other approaches.

## 3. Hypotheses Development

This section describes the proposed causal model to predict young consumers' intention to participate in the sharing economy through platforms. The suggestions mentioned in the fourth approach of the previous section have been adopted in this study. They refer to the need to include new variables and relationships in models jointly with others that have been previously verified. Similar to the intrinsic and extrinsic motivation approach, the proposed model includes internal and external consumer variables. In the literature on consumer behavior, there are some similar models [84]. However, it should be noted that the relationship between reputation, trust, and satisfaction, or between social norm, attitude, and intention, which are included in this study, are novel in the literature [85]. Finally, regarding the starting variables of the new causal models, values [78] and trust stand out as central elements [86].

### 3.1. The Importance of Values

The need to incorporate values in this type of study is critical because values affect preferences, expectations, and behavior of individuals from different generations, as in the case of Generation Z [78,82]. Values in this study mean the extent to which a person has a favorable opinion and judgment of the sharing economy and what it represents [82,87,88]. Regarding consequences, it has been specifically proven that values related to sustainability and solidarity affect consumers' preferences and expectations associated with Electronic Word of Mouth (eWOM) [89–92]. EWOM is defined as the information exchanges, evaluations and comments made through sites [93]. Taking into account the above, and the social and technological nature of young consumers and e-commerce platforms, the first hypothesis states that:

**Hypothesis 1 (H1).** *Young consumers' values related to the sharing economy in tourism have a positive effect on their perceptions about the importance of eWOM communication through platforms.*

*3.2. The Effect of eWOM Communication on the Perceived Quality of Information about Products and Service*

Quality information about products and services is essential in the intangible and online context of the sharing economy in tourism. It helps reduce perceived risk, promotes reputation, trust, and encourages participation [59,94,95]. Online information about the quality of the product or service must be current, accurate, relevant, useful, and complete [96,97]. One of the factors that most strongly influences the quality of the consumers' perceived information about the product and the service is eWOM communication [98,99]. This communication is defined as the evaluation, comments, recommendations, and opinions developed online by users and consumers [66,100]. In addition, young consumers rely more on eWOM communication than on face-to-face communication or official online information from the provider [101–103]. Therefore, the second hypothesis states that:

**Hypothesis 2 (H2).** *In the sharing economy in tourism, eWOM communication has a direct and positive effect on young consumers' perceptions of quality information than the platform offers on products and service.*

*3.3. The Effect of Quality Information about Products and Service on Reputation*

In the sharing economy in tourism, reputation represents a synthesis of consumers' opinions, perceptions and attitudes regarding the reliability and credibility of a tourism company [14,43,94,104]. Reputation is particularly important in the online context of the sharing economy because the quality of products and services is difficult to verify when they are purchased [16,105]. Therefore, reputation depends on the quality of online information about products and services from a company, among other stakeholders [106–108]. For this reason, some researchers include in their causal models to predict intention, information about products and services as an independent variable [109]. Moreover, suppliers also make efforts to supply optimal levels of quality information about products and services on sharing economy platforms in order to maximize profitability [59,110]. Thus, the third hypothesis holds:

**Hypothesis 3 (H3).** *In the sharing economy in tourism, young consumers' perceived quality of information about products and services has a direct and positive effect on company reputation.*

*3.4. The Effect of Reputation on Trust*

Trust is also a fundamental variable in the online and insecure context of commerce and tourism carried out by young consumers through shared economy platforms [111,112]. Trust is a subjective feeling about how the supplier will deliver as promised [113,114]. More specifically, trust refers to the positive expectations of the consumer about the supplier's competency, (effective professional), benevolent (good behavior) and honest (integrity) conduct through an online platform [95,115]. Regarding its antecedent, it has been found in the literature that positive reputation is the variable that most influences trust [80,116,117]. People would not buy travel products and services if they did not trust the service provider, and they would not trust it if it had a bad reputation [118,119]. Based on the above, the following hypothesis states that:

**Hypothesis 4 (H4).** *In the sharing economy in tourism, the reputation of a company has a direct and positive effect on young consumers' trust.*

*3.5. The Effect of Trust on Satisfaction*

Satisfaction plays a vital role in relationships with customers, and it is one of the variables that most influences intention to participate in the tourism sharing economy [2,120,121]. It has been shown through different approaches that in the sharing economy, trust is an

antecedent of consumers' satisfaction [19,80,122]. Both constructs are linked through expectations to consumers' behavior and their real experience [103,123]. On the one hand, trust is related to consumers' perceptions that pre-purchase conditions are present to facilitate the success of the transaction, anticipating the fulfillment of these expectations [103,124]. On the other hand, satisfaction is defined as the overall and cumulative perception of the degree to which a buyer's previous expectations are confirmed after an online purchase [120]. Therefore, as consumer trust is based on the a priori belief in the possibility of obtaining a satisfactory result in a transaction, it is logical to think that trust facilitates satisfaction [125,126]. Therefore, the following hypothesis states that:

**Hypothesis 5 (H5).** *In the sharing economy in tourism, trust has a direct and positive effect on young consumers' satisfaction.*

### 3.6. The Effect of Satisfaction on Social Norm

A social norm is formed by a socialization process, and it represents a person's beliefs about how other important and significant people expect them to behave [37,87]. In this way, social norm influences behavior because, according to the principle of compliance, which has been empirically verified, people tend to adjust their behavior to expectations of significant people [88,127,128]. In particular, social norm is influenced by satisfaction through expectations. Satisfaction implies the fulfillment of a person's expectations, which must be in tune with the expectations of significant others, in turn reinforcing the social norm [18,58,63,129]. In addition, existing studies have confirmed that in the online context, there is pressure toward conformity and a tendency to imitate other people [130]. Furthermore, the causal relationship between satisfaction and social norm is more significant within the same generational segment [26]. For these reasons, the following hypothesis states that:

**Hypothesis 6 (H6).** *In the sharing economy in tourism, young consumers' satisfaction has a direct and positive effect on social norm.*

### 3.7. The Influence of Social Norm on Attitude

Attitude refers to the favorable or unfavorable evaluation of a person about an object, a subject or behavior. It is a critical driver in predictive models developed for studying the antecedents of consumers' behavioral intentions [19]. Regarding antecedents, attitude depends on social norm, i.e., individuals have a favorable attitude to an object (e.g., sharing economy) when it is approved and valued positively by others [51,131]. In the sharing economy, the influence of social norms on attitude is especially pronounced, as these platforms are community-based and rely heavily on word of mouth marketing [66,96,132]. This influence is more significant in the case of Generation Z, a generation that thinks collectively and shares attitudes, values, and perceptions [25,26]. Therefore, the following hypothesis states that:

**Hypothesis 7 (H7).** *In the sharing economy in tourism, social norm has a direct and positive effect on young consumers' attitude.*

### 3.8. The Effect of Attitude on Intention to Participate

Various authors have verified in different sectors and through different theories (e.g., Theory of Planned Behavior) the positive and direct influence of attitude on the intention to participate [19,133,134]. This relationship has also been confirmed in the sharing economy in tourism [18,87,135]. However, due to the difficulties of having real data about consumers' actual participation, this is usually measured through intentions, since intention has been shown to be the best predictor of behavior in tourism [10,11,136]. Therefore, the following hypothesis specifies that:

**Hypothesis 8 (H8).** *Young consumers' attitude toward the sharing economy in tourism has a positive effect on their intention to participate.*

Therefore, according to the above, the model proposed in this study is as follows (Figure 1):

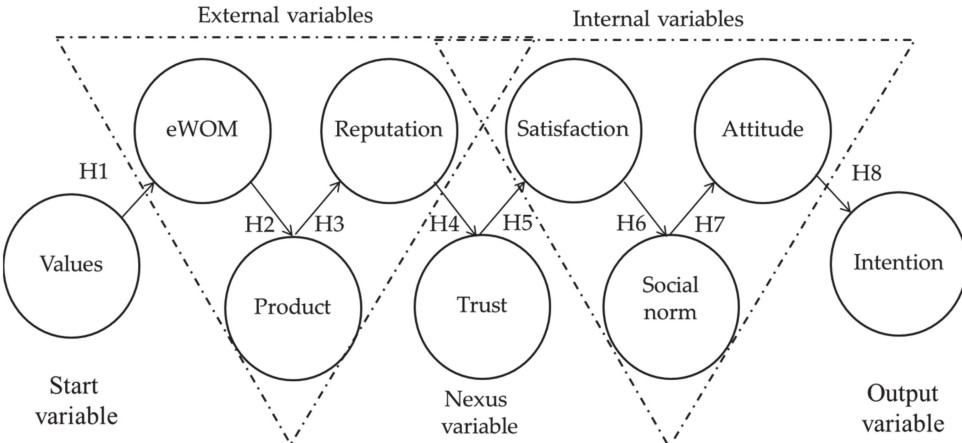

**Figure 1.** Research proposed model. Source: Authors.

## 4. Research Methodology

### 4.1. Questionnaire Development

In this study, we examine young consumers' intention to participate in the sharing economy in tourism. For data collection, a questionnaire was designed in three steps and was applied online in the first week of May 2020. First, a literature analysis was conducted to generate content validity with the collaboration of two experts to design the measures and generate content validity [137]. In particular, the following studies were consulted: Gupta et al. [78] (items related to values), Kwok and Xie [138] (items related to eWOM and the commercial proposal of the platform), Wu et al. [103] (reputation, trust, and satisfaction), Weng et al. [2] and Xu [121] (satisfaction), Mahadevan [87] (social norm and attitude), and Yen and Teng [136] (intention to participate). After a pretest, the initial questionnaire included 22 items using a Likert scale with five response alternatives (1: no agreement to 5: total agreement).

Second, according to Šegota, Mihalič, and Kuščer [139], the Delphi technique was used online through two rounds involving two groups via Google Meet to design the initial contents of the items. The two groups were made up of three and four professors who are specialists in the sharing economy in tourism and teach in the Degree in Tourism. The groups were led by the authors of this study. After a pretest and considering the principles of brevity and simplicity to reduce methodological costs [140,141], the initial Likert scale included 21 items. Each item included five response alternatives (1: no agreement to 5: total agreement) (Table 1). Third, to identify the latent variables, an exploratory factor analysis was carried out. After various analyses, a structure of nine latent variables was obtained. These variables are consumer values about sharing economy (VAL), possibility of using eWOM (eWOM), quality information about products and services (PRO), company reputation (REP), consumer trust (TRU), consumer satisfaction (SAT), social norm (SOC), and consumer attitude toward tourism (ATT). Consumers' intention to participate in the sharing economy (INT) is the dependent variable. Through this factor analysis, four items were eliminated corresponding to the latent variables VAL (1), PRO (2), and SOC (2). Being reflective items and since all indicators of a construct are interchangeable, the nature of the latent variables was not significantly altered. Additionally, the results of the factor analysis verified that the overall reliability of the questionnaire increased by eliminating these items, since Cronbach's alpha changed from 0.81 to 0.86. The existence of two items in all latent variables was accepted because the items have a reduced correlation with other items, and

the correlation between them was more significant than 0.70 [142]. A control item was also included related to the degree of consumer participation in the shared economy in tourism. It is important to note that the subjects were asked about their real and future participation in the shared economy in tourism, either through some platforms or others (e.g., Airbnb, Uber, BlaBlaCar).

**Table 1.** Details of the Sample.

| Year | Total | % | Male | %/532 | Female | %/532 |
|---|---|---|---|---|---|---|
| 1st | 139 | 26.13% | 53 | 9.96% | 86 | 16.17% |
| 2nd | 140 | 26.32% | 47 | 8.83% | 93 | 17.48% |
| 3rd | 124 | 23.30% | 46 | 8.65% | 78 | 14.66% |
| 4th | 129 | 24.25% | 49 | 9.21% | 80 | 15.04% |
| Total | 532 | 100.00% | 195 | 36.65% | 337 | 63.35% |

*4.2. Data Collection and Sample Profile*

The sample was selected randomly in the second week of May 2020 after completing a cluster study by degree, academic year, and gender. It should be noted that the authors had access to the names of the students, as well as their email addresses, gender, degree, and year of study. Thus, the sample was stratified by degree, academic year, and gender. Students were included in the sample in a representative way, including students from the Tourism degree. The university context selected for this study is the Canary Islands (La Laguna University), one of the most important tourism destinations in Spain. Only one university was selected for data collection. Although the generational debate has not been closed, this study adopts that approach according to which young consumers share globally similar perceptions and behavior [143,144]. Moreover, it was found that university students represent the Generation Z well [29,30], and they are an adequate representation of virtual consumers [32]. Additionally, the size effect (0.15) and power (0.90) indicators were specified at a 95% confidence level [145]. The sample initially included 556 subjects (37.59% men and 62.41% women; mean age of the respondents was 20.89 years old), an appropriate size when structural equations are used, since it is higher than 200 subjects [146], and it agrees with the "ten-time rule" [147]. Data collection was carried out online through Google forms. The online mode was selected because people belonging to Generation Z accept this technological context and because it was known that the respondents had online access. Although the questionnaire had been prepared with care, interviewers explained the sharing economy, questionnaire's content, and instructions, and they answered questions from respondents through Google meet. As the survey was associated with academic work, educational incentives were given to students to maximize participation.

*4.3. Data Analysis*

Data were first examined in a descriptive way through SPSS 22 to obtain the total scores, percentages, means, standard deviations, skewness, and kurtosis. Then, to test the hypotheses, the Partial Least Squares Structural Equation Modeling approach (PLS-SEM) was applied through SmartPLS-3 software (3.3.2 version). We selected PLS for its potential to explain the theory [148], and its great predictive potential of human behavior [149]. Moreover, in this approach, it is not necessary to assume a normal distribution of data [150]. The most recent guidelines in the application of PLS-SEM in tourism research have been followed, including an Importance-Performance Analysis (IPMA) [147,151].

## 5. Findings

*5.1. Sampling Data Results*

The sample initially included 556 subjects. After responses with missing data (n = 10, 6 men and 4 women) and outliers (n = 14.8 men and 6 women) were excluded, a total of 532 (N = 532) respondents were included in the final sample, with an overall response of 95.68%. This elimination did not significantly affect the representativeness of the sample in

terms of degree, academic year, or gender. Additionally, the percentage of men (36.65%) and women (63.35%) in the sample is similar to that existing in the selected university and the target population (see Table 1). Finally, the mean age of the respondents was 20.97 years old.

*5.2. Descriptive Data*

As shown in Table 2, young consumers scored high on all the items with 88.88% of items obtaining a valuation above 70%. The items with the lowest score are related to intention (INT1 and INT2) (68.61% and 63.12% respectively), and those with the highest scores were those of trust (TRU) and satisfaction (SAT) (97.57% and 94.47% respectively). Moreover, young people state that they have used sharing economy platforms at a medium level (CONT = 50.36%). On the other hand, eleven of the eighteen items obtained a minimum score of two or three points. In addition, although PLS does not require normality in data distribution, the results showed the existence of relative normality. Most skewness and kurtosis values were in absolute terms around 2 and 7, respectively, which are limits considered adequate for samples higher than 300 subjects [148]. Additionally, since the standard deviation of all items was less than half of the mean, it can be affirmed that there were no extreme balances or values.

**Table 2.** Descriptive data.

| Item | Min/Max | % ≥50% | MD ≥3 | SD ≤1.5 | Skew. ≤\|2\| | Kurt. ≤\|7\| |
|---|---|---|---|---|---|---|
| **Consumer values about tourism sharing economy (VAL)** | | | | | | |
| 1. I like the sharing economy in tourism because it favors sustainability | 1/5 | 72.29% | 3.59 | 1.00 | −0.65 | −0.07 |
| 2. I like the sharing economy in tourism because it favors solidarity | 1/5 | 74.47% | 3.81 | 1.00 | −0.44 | −0.48 |
| **E−WOM communication (WOM)** | | | | | | |
| 1. I like to read eWOM opinions and ratings on the platform | 2/5 | 79.40% | 3.99 | 0.81 | −0.21 | −1.11 |
| 2. I like to write eWOM opinions and ratings on the platform | 1/5 | 71.39% | 3.63 | 1.04 | −0.36 | −0.35 |
| **Quality information about product and service (PRO)** | | | | | | |
| 1. I value precise and current information about products and services | 3/5 | 89.25% | 4.46 | 0.67 | −0.88 | −0.43 |
| 2. I value relevant and useful information about products and services | 3/5 | 92.63% | 4.73 | 0.57 | −1.12 | 0.25 |
| **Reputation of the supplier−company (REP)** | | | | | | |
| 1. The company must be competent and professional | 2/5 | 86.77% | 4.38 | 0.72 | −0.83 | 0.61 |
| 2. The company must fulfill what it has promised | 2/5 | 83.68% | 4.16 | 0.80 | −0.68 | −0.20 |
| **Consumer trust (TRU)** | | | | | | |
| 1. I value the suppliers' competence and professionalism | 2/5 | 93.83% | 4.73 | 0.69 | −2.17 | 4.16 |
| 2. I value suppliers' honesty and integrity | 3/5 | 94.47% | 4.76 | 0.61 | −1.62 | 1.73 |
| **Consumer satisfaction (SAT)** | | | | | | |
| 1. In the tourism sharing economy, I want to feel satisfied with the purchase | 3/5 | 97.57% | 4.85 | 0.39 | −3.08 | 9.38 |
| 2. I wish to see my expectations fulfilled by the purchase | 3/5 | 94.47% | 4.82 | 0.54 | −1.83 | 2.41 |
| **Social norm (SOC)** | | | | | | |
| 1. My friends would accept that I used the sharing economy in tourism | 2/5 | 78.16% | 3.96 | 0.93 | −0.31 | −0.87 |
| 2. My family would accept that I used the sharing economy in tourism | 1/5 | 70.53% | 3.53 | 1.05 | −0.45 | −0.45 |
| **Consumer attitude toward tourism (ATT)** | | | | | | |
| 1. The sharing economy is good and beneficial for tourism | 2/5 | 74.55% | 3.71 | 0.94 | −0.13 | −0.82 |
| 2. I am in favor of the sharing economy in tourism | 1/5 | 74.74% | 3.73 | 0.93 | −0.34 | −0.34 |
| **Consumer intention to participate (INT)** | | | | | | |
| 1. I intend to use the sharing economy in tourism | 1/5 | 68.61% | 3.45 | 1.03 | −0.31 | −0.49 |
| 2. As soon as I can, I will use a sharing economy platform in tourism | 1/5 | 63.12% | 3.19 | 1.01 | −0.10 | −0.67 |
| **Control item (CONT)** | | | | | | |
| I have used a sharing economy platform in tourism | 1/5 | 50.36% | 2.51 | 0.90 | −0.82 | −0.47 |

### 5.3. Assessment of the Overall Model

The values of SRMR (Standardized root mean square residual) as an approximate model fit for PLS−SEM were calculated. SRMR assesses the average magnitude of the discrepancies between observed and expected correlations. The results revealed an SRMR model fit value of 0.069, which is considered adequate in a more conservative version because it is less than 0.08 [152]. The common method bias (CMB) was examined through PLS−SEM. CMB is the spurious variance that is attributable to the measurement method. In this study, the model was considered free of CMB because all variance inflation factors (VIF) resulting from a full collinearity test were lower than 3.3, both with respect to the indicators and the constructs or latent variables [153].

### 5.4. Test of the Measurement Model

The study of the measurement model with reflective indicators verifies reliability (individual and compound) and validity (convergent and discriminant) [146]. Regarding individual and composite reliability (CR), results showed that all values reached the required level ($\lambda \geq 0.70$; CR $\geq 0.70$) [151] (Table 3). Therefore, all the observed variables were measuring their corresponding latent variable and the measurement model was internally consistent [147]. Regarding validity, the values of the average variance extracted (AVE) showed the amount of variance attributed to the constructs. AVE was in all cases greater than 0.50 (AVE > 0.50) (Table 3), thus confirming the convergent validity of the model [147].

**Table 3.** Measure Model: Basic Data.

| Construct | Items | Loading λ >0.70 | CR >0.70 | AVE >0.50 | $R^2$ >0.50 | $Q^2$ >0 |
|---|---|---|---|---|---|---|
| Consumer values about the sharing economy (VAL) | VAL1<br>VAL2 | 0.919<br>0.887 | 0.897 | 0.813 | − − − | − − − |
| E−WOM communication (WOM) | WOM1<br>WOM2 | 0.844<br>0.899 | 0.863 | 0.758 | 0.270 | 0.199 |
| Quality information about product and service (PRO) | PRO1<br>PRO2 | 0.821<br>0.772 | 0.772 | 0.627 | 0.197 | 0.123 |
| Reputation of the supplier−company (REP) | REP1<br>REP2 | 0.935<br>0.893 | 0.911 | 0.834 | 0.142 | 0.114 |
| Consumer trust (TRU) | TRU1<br>TRU2 | 0.823<br>0.918 | 0.864 | 0.774 | 0.272 | 0.195 |
| Consumer satisfaction (SAT) | SAT1<br>SAT2 | 0.850<br>0.762 | 0.783 | 0.645 | 0.183 | 0.116 |
| Social norm (SOC) | SOC1<br>SOC2 | 0.911<br>0.948 | 0.927 | 0.864 | 0.117 | 0.098 |
| Consumer attitude toward tourism (ATT) | ATT1<br>ATT2 | 0.956<br>0.937 | 0.946 | 0.895 | 0.321 | 0.275 |
| Consumer intention to participate (INT) | INT1<br>INT2 | 0.899<br>0.938 | 0.914 | 0.841 | 0.524 | 0.292 |

To analyse discriminant validity, it was verified that the square root of the average variance extracted (AVE) of each variable (data in bold in Table 4) was greater than the variance shared with the other variables (Criteria of Fornell and Larcker) [154]. Next, the discriminant validity was also verified through the Heterotrait−Monotrait Ratio (HTMT) (values above the diagonal in Table 4). Regarding HTMT rate, all values were lower than 0.85 in all relationships [147]. Therefore, the results show that the measurement model had acceptable convergent and discriminant validity.

### 5.5. Test of the Structural Model

The structural model shows the relationship between the latent variables [149] (Table 5). First, it was found that all relationships had positive signs, as do their corresponding

hypothesis. Regarding the magnitude of the causal relationships, it was verified that the path coefficients (β) (standardized regression weights) reached, in all cases, levels above 0.300, which is the optimum level (β ≥ 0.300) [149]. As Table 5 and Figure 2 show, causal relationships with greater weight were obtained between attitude (ATT) and intention to participate (INT) ($β_{H8}$ = 0.601, t-value = 17.875, $p$ < 0.001), between social norm (SOC) and attitude (ATT) ($β_{H7}$ = 0.564, t-value = 19.300, $p$ < 0.001), as well as between values (VAL) and eWOM (WOM) ($β_{H1}$ = 0.524, t-value = 19.948, $p$ < 0.001), and also between reputation (REP) and trust (TRU) ($β_{H4}$ = 0.521, t-value = 18.273, $p$ < 0.001). The weakest relationship was between commercial product (PRO) and reputation (REP) ($β_{H3}$ = 0.375, t-value = 10.628, $p$ < 0.001), and between satisfaction (SAT) and social norm (SOC) ($β_{H6}$ = 0.341, t-value = 9.567, $p$ < 0.001). The bootstrapping analysis carried out with 5000 sub−samples and 500 cases revealed that all relationships obtained a high significance [147]. Therefore, all hypotheses were accepted.

**Table 4.** Discriminant Validity: Criteria of Fornell and Larcker and HTMT rate.

| Construct | VAL | WOM | PRO | REP | TRU | SAT | SOC | ATT | INT |
|---|---|---|---|---|---|---|---|---|---|
| VAL | 0.904 | 0.700 | 0.614 | 0.370 | 0.294 | 0.435 | 0.276 | 0.171 | 0.105 |
| WOM | 0.520 | 0.875 | 0.844 | 0.455 | 0.188 | 0.292 | 0.135 | 0.155 | 0.145 |
| PRO | 0.329 | 0.444 | 0.794 | 0.640 | 0.811 | 0.842 | 0.439 | 0.167 | 0.308 |
| REP | 0.296 | 0.345 | 0.377 | 0.915 | 0.662 | 0.502 | 0.236 | 0.176 | 0.096 |
| TRU | 0.239 | 0.112 | 0.443 | 0.521 | 0.873 | 0.759 | 0.481 | 0.097 | 0.202 |
| SAT | 0.248 | 0.175 | 0.477 | 0.297 | 0.428 | 0.805 | 0.543 | 0.215 | 0.071 |
| SOC | 0.236 | 0.080 | 0.245 | 0.192 | 0.374 | 0.341 | 0.929 | 0.633 | 0.715 |
| ATT | 0.141 | 0.103 | 0.003 | −0.083 | 0.081 | 0.141 | 0.560 | 0.947 | 0.690 |
| INT | 0.086 | 0.060 | 0.156 | −0.040 | 0.159 | 0.025 | 0.597 | 0.595 | 0.917 |

Note: VAL: Values about sharing economy; WOM: e−WOM communication; PRO: Quality information about product and service; REP: Reputation of the supplier−company; TRU: Consumer trust; SAT: Consumer satisfaction; SCO: Social norm; ATT: Consumer attitude toward tourism; INT: Consumer intention to participate.

**Table 5.** Direct Effects, Significance and Confirmation of Hypotheses.

| Hypothesis | Path Coefficient (β) | t-Value | $p$ | Supp. |
|---|---|---|---|---|
| H1: VAL → WOM | 0.524 | 16.948 | 0.000 | YES |
| H2: WOM → PRO | 0.438 | 11.732 | 0.000 | YES |
| H3: PRO → REP | 0.375 | 10.628 | 0.000 | YES |
| H4: REP → TRU | 0.521 | 18.273 | 0.000 | YES |
| H5: TRU → SAT | 0.428 | 11.638 | 0.000 | YES |
| H6: SAT → SOC | 0.341 | 9.567 | 0.000 | YES |
| H7: SOC → ATT | 0.564 | 19.300 | 0.000 | YES |
| H8: ATT → INT | 0.601 | 17.875 | 0.000 | YES |

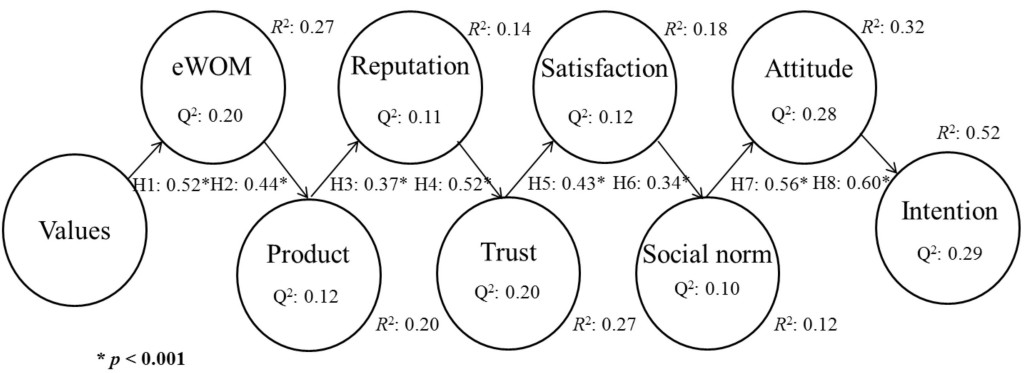

**Figure 2.** Structural model. Source: Authors.

*5.6. Analysis of the Predictive Validity of the Model*

Regarding the predictive validity of the proposed model, the $R^2$ indicator (coefficient of determination) was calculated within the sample. The value of $R^2$ relative to the dependent construct (INT) was 0.524 ($R^2$ = 0.524), with 0.500 being the minimum acceptable value (Table 2) [155]. In addition, indicator $Q^2$ was calculated in a redundancy−based prediction way and through the blindfolding process. It showed all values above zero ($Q^2 \geq 0$) and, in all cases were in the interval (0.10, 0.25), both about the items and the latent variables [147]. Therefore, we confirm that the proposed model has satisfactory structural properties and sufficient predictive potential. Through PLSPredict, it was found that the predictive potential of the model is high since all the dependent construct indicators produced lower prediction errors using RMSE (root mean squared error) compared to the LM (linear regression model) [155]. Finally, the finite mixture PLS approach revealed that the results were not distorted by unobserved heterogeneity [155].

*5.7. Importance−Performance Analysis*

To determine the latent variables to prioritize, an Importance−Performance Analysis (IPMA) was carried out. IPMA method was developed by Martilla and James [156], and it compares the antecedent constructs' importance in shaping a certain target construct (through the total effects), with their performance (through their average scores) [157–159]. The results showed (Figure 3) that in the model there were no variables with reduced importance and reduced performance. Attitude (ATT) and social norm (SOC) are the most relevant variables of the model because of their high importance and high performance in relation to participation intention in the tourism sharing economy. Both variables are near the maximum productivity line and require priority attention in terms of resources and time [160].

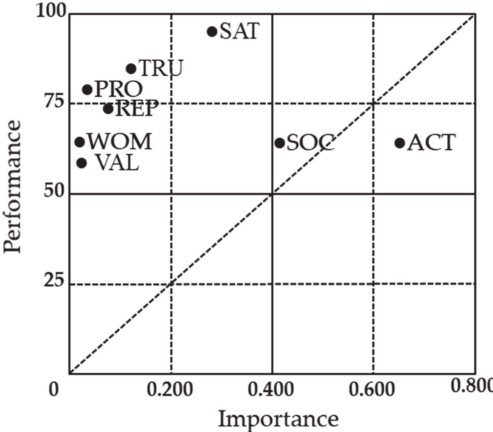

**Figure 3.** Importance−Performance Analysis (IPMA). Source: Authors.

## 6. Discussion

This study aims to understand better young consumers' intention to participate in the tourism sharing economy through the most popular platforms (e.g., Airbnb, Uber, BlaBlaCar). The suggestions of other authors regarding complementing existing theories with different variables and relationships have been adopted. In particular, the theoretical and practical gaps and limitations of the Social Theory of Exchange (SET) to explain intention to participate in the sharing economy have been accounted for in this study. SET has been used as well as including other internal and external variables and relationships in the model, as other authors have proposed. These variables and relationships have not previously been studied together. In the sections below, the theoretical and practical implications are discussed.

### 6.1. Theoretical Implications

Regarding responses to items, it can be confirmed that young people give importance to all the variables presented in this study. This includes the intention to participate in the tourism sharing economy, trust, and satisfaction. This result confirms the findings of other authors about young consumers' attitudes in favor of participating in the sharing economy [161]. On the other hand, it should be noted the moderate levels were reached by two items related to the intention to participate (INT1 = 68.61%, INT2 = 63.12%) and the control item (CONT = 50.36%), which may be due to the novelty of sharing economy and the age of the people of the sample.

An explanatory model of the intention to participate has been generated, which explains more than 50% of the variance of intention. The proposed causal model is complete concerning the number of dimensions (8), demonstrating the complexity of consumer behavior [18,47]. It addresses the concerns of other researchers regarding the development of more sophisticated causal models due to limitations of existing theories (e.g., SET). Moreover, despite the number of dimensions in the model, some of its partial relationships have already been confirmed by other authors. The model is practical to understand, predict, and it can be used to encourage young consumers' participation in the sharing economy. All the hypotheses of the proposed model have been confirmed. Therefore, the chain of direct and indirect effects of the model has not been broken, and this makes it possible to significantly predict the intention to participate in the tourism sharing economy.

In particular, the model shows the importance of values that trigger the chain of effects of young consumers' intention to participate in the tourism sharing economy. This is supported by other authors who have emphasized that values are at the beginning of any behavior [87]. Values influence eWOM importance ($\beta_{H1} = 0.524$, t-value = 16.948, $p < 0.001$) and indirectly participation behavior. This confirms the relevance that this type of communication has for young people, especially in the online, intangible, and uncertain context of the sharing economy [162]. It is worth noting that after values, the first group of three dimensions (eWOM, reputation and product) refer to the external environment of the consumer. These dimensions influence through trust another group of three dimensions (satisfaction, social norm, and attitude), which correspond to consumers' internal or personal sphere. Thus, in the model, trust is the central dimension, which serves as a link between the three external dimensions (eWOM, product and reputation) and the three internal ones (satisfaction, social norm, and attitude). It confirms the central importance that trust has in purchasing intention and behavior in the sharing economy and e−commerce [125].

Indeed, the importance that, according to other authors, young people give to eWOM communication is confirmed. For Generation Z, it is a way to obtain quality information about products and services in the online context of the sharing economy in tourism ($\beta_{H2} = 0.438$, t-value = 11.732, $p < 0.001$). This communication is a primary, permanent, real-time and up-to-date source of information that allows young consumers to know the characteristics of products and services and to reduce risks and uncertainty [108]. Likewise, any textual, photographic, or video information about a company's products and services influences the reputation of the supplier−company ($\beta_{H3} = 0.375$, t-value = 10.628, $p < 0.001$) [16].

The findings of other researchers about the influence of reputation on consumers' trust have also been confirmed in this study ($\beta_{H4} = 0.521$, t-value = 18.273, $p < 0.001$) [117]. This result confirms the key rule that people would not buy tourism products and services if they did not trust the provider or if it had a bad reputation [163]. It also shows that trust influences satisfaction in a significant way ($\beta_{H5} = 0.428$, t-value = 11.638, $p < 0.001$) and that both variables focus on expectations, before and after the transaction respectively [126,128]. Therefore, based on values, consumers' external variables (eWOM, product and reputation) are responsible for generating previous expectations associated with trust. It is a critical aspect, especially as sharing economy platforms and e−commerce consumers have a reduced tolerance of risk [164].

At the same time, it is confirmed that the fulfillment of consumers' previous expectations (satisfaction) reinforces the social norm ($\beta_{H6}$ = 0.341, t-value = 9.567, $p < 0.001$). The expectations that others have about the consumer confirm the consumer's expectations and the expectations of peers and friends, since this previous behavior has been carried out because of the influence of the social norm itself [131]. Therefore, in the online context, there is a pressure to conform to others and efforts are made to imitate them [130].

We also confirmed the findings of other authors regarding the importance of attitude, its antecedents, and consequences, obtained in other sectors and with different samples [51,133]. In particular, it has been found that attitude is significantly influenced by social norm ($\beta_{H7}$ = 0.564, t-value = 19.300, $p < 0.001$) and directly influences the intention to participate in tourism sharing economy platforms ($\beta_{H8}$ = 0.601, t-value = 17.875, $p < 0.001$) [87]. Finally, according to the results of the IPMA analysis, attitude, and social norms are relevant variables to influence the intention to participate in the tourism sharing economy. The question now arises as to whether these two variables and the others can be controlled or influenced by companies.

*6.2. Practical Implications*

In addition to the theoretical aspects exposed in the previous sections, the results obtained in this study provide practical implications for tourism companies and other institutions (e.g., local government, education institutions) for managing sharing economy platforms in tourism and increasing consumers' participation in these platforms. This, in turn, will make it easier to guide professionals in their market orientation and development of appropriate and effective communication and messages.

First, the descriptive results show that the attitudes and perceptions of young people are favorable to the sharing economy in tourism. However, their current participation and intention to participate can be considered average. It indicates that young people have an excellent predisposition and attitude toward shared exchanges and have the potential to participate. Moreover, the literature analysis and the descriptive results show that the profile of Generation Z is suitable to promote their participation in the sharing economy. This information is useful for business marketing management.

The proposed model includes dimensions and relationships that lead significantly to the participation of young people on tourism sharing platforms. The number and diversity of variables included in the model allow us to verify the complexity of consumer behavior in the context of the tourism sharing economy. This confirms that young consumers' intention to participate in sharing depends on a process in which consumers' internal and external aspects are present. This is a reality to be taken into account by marketing professionals. However, not all dimensions of the model are equally relevant and manageable by companies (see IPMA analysis).

In some cases, the variables of the model are approachable through other contexts, such as local government and education. This is the case for consumers' attitude about the sharing economy and social norms. It is also true for the sharing values that in the proposed model start the chain of effects that leads to participation. They are abstract variables, although associated with a group of significant people belonging to the consumer's social context. The messages and actions of the companies can promote these variables, and they are also manageable in online education and among families during a crisis and after it. Specifically, the actions of companies must go in the same direction and generate synergies regarding these variables, and not contradict them.

According to the results, and as some authors have proposed [165], companies that manage sharing economy platforms in tourism should facilitate eWOM communication in a market orientation context. This is essential in the case of Generation Z, who is a very friendly, communicative, and technological segment. Likewise, in the market orientation context, information (textual, images, videos) about products and service must be presented on the platforms based on eWOM communication and segment profile. To do this, companies must organize and develop business intelligence to previously identify

consumers' needs, expectations, and desires, and then satisfy them. These actions would favor companies' reputations and provide sufficient trust in terms of expectations before purchase. Companies must assume that consumer trust is the central variable of the proposed model and, ultimately, of online shopping behavior.

Satisfaction is, with trust, another fundamental variable of electronic commerce that is the link to the satisfaction of expectations, a central element of market orientation. The social reinforcement derived from satisfaction directly and positively influences the attitudes of young people toward the sharing economy, thereby further encouraging their intention and behavior to participate. Therefore, satisfying consumer expectations is a way to reinforce the social norm. Communication from the government and tourism institutions should also support the development of positive attitudes and expectations.

As the IPMA analysis showed, attitude and social norm are fundamental variables to improve intention. Additionally, marketing professionals of sharing economy tourism companies should be aware that trust, reputation, and satisfaction are central variables of the process that culminates in the intention to participate. Therefore, depending on the results and the proposed causal model, companies can improve young consumers' intention to participate in the sharing economy through market orientation, taking into account the following sequence or chain: values, trust and intention to participate. These are internal variables but given their relevance in electronic commerce and their role as a dependent variable (intention) they have been strategically placed in the model.

$$\text{Values} \rightarrow 3 \text{ External variables} \rightarrow \text{Trust} \rightarrow 3 \text{ Internal variables} \rightarrow \text{Intention}$$

Finally, the generational approach adopted in this study allows companies of the tourism sharing economy to consider the possibility of carrying out global actions in order to encourage the participation of young people in the platforms.

## 7. Conclusions

The sharing economy has become a new consumption model with high growth and potential, especially in tourism and after the COVID−19 Pandemic [166]. This study takes into account that consumer participation is decisive in this consumption model. However, its study has been relegated to the background except for studies that emphasize the importance of its economic, legislative, or structural aspects. In addition, the literature and the results of this study confirm the potential of young consumers to participate in the tourism sharing economy. It has also been shown in this study that consumer behavior in this field is complex and depends on a multitude of variables, particularly on intention. Moreover, there are limitations and insufficiencies with existing theories and models to explain and predict the intention to participate in the tourism sharing economy. In this sense, several authors suggest combining the Social Exchange Theory (SET) with other variables and relationships that have been previously tested. These suggestions have been addressed in this study. The result is an integrated, complete, useful, and realistic model that explains more than 50% of young consumers' intention to participate in the tourism sharing economy. The model includes consumers' perceptions of internal and external variables that have not previously been studied jointly.

Although our study contributes to improving theoretical and practical knowledge about the intention to participate in the tourism sharing economy, it is not without limitations. The main limitation of this work is related to the selection and combination of the internal and external variables included in the proposed model, given the great diversity of variables in the literature when studying online purchase intention. Access to the sample has also been complicated. Additionally, the study has been limited to a single geographic context and a single University. As for future research lines, it may be of interest to include other variables in the model or to study the validity of the model for the same segment in other universities, geographical contexts, and sectors in order to verify the generational approach that was adopted. Moreover, the global nature of the results of generational studies with young consumers (e.g., Millennials and Generation Z) is not

entirely conclusive, particularly in the case of tourism. A future line of research could delve into these aspects. Additionally, it would be interesting to study the model by selecting some specific tourism sharing platforms for a more in−depth analysis of the sector in a disaggregated manner. Finally, it must be considered the impact caused on tourism by COVID−19 [166,167] and that sharing economy activities may be in a situation of certain precariousness [168]. Therefore, a future research line could aim to carry out the same study when the Pandemic has passed, and tourism has returned to normal.

**Author Contributions:** J.A.M.-G., E.P.-L. and A.B.-B. were involved in the conceptualization, literature review, methodology design, investigation, data analysis, and review. Writing, J.A.M.-G., A.B.-B. All authors have read and agreed to the published version of the manuscript.

**Funding:** This research received no external funding.

**Institutional Review Board Statement:** This study did not require ethical approval.

**Informed Consent Statement:** Informed consent was obtained from all subjects involved in the study.

**Data Availability Statement:** The data presented in this study are available on request from the corresponding author. The data are not publicly available due to restrictions of privacy.

**Conflicts of Interest:** The authors declare no conflict of interest.

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
