# Peer review of "Young Consumers’ Intention to Participate in the Sharing Economy: An Integrated Model"

_sustainability, doi:10.3390/su13010430_

Round 1

Reviewer 1 Report

Dear authors,

Congratulations for your paper!

Author Response

Thank you very much.

Reviewer 2 Report

The authors of the manuscript presented the topic of the sharing economy in tourism. In the manuscript the authors presented a very important point, although the manuscript has some drawbacks that should be corrected before publication. The topic presented in the manuscript is interesting.

Please write in conclusions about the current crisis in tourism caused by the COVID-19 pandemic (including the sharing economy in tourism). Suggested publications:

  • Roman, M.; NiedzióĹ‚ka, A.; KrasnodÄ™bski, A. Respondents’ Involvement in Tourist Activities at the Time of the COVID-19 Pandemic. Sustainability 2020, 12, 9610. https://doi.org/10.3390/su12229610
  • Fermani, A.; Sergi, M.R.; Carrieri, A.; Crespi, I.; Picconi, L.; Saggino, A. Sustainable Tourism and Facilities Preferences: The Sustainable Tourist Stay Scale (STSS) Validation. Sustainability 2020, 12, 9767. https://doi.org/10.3390/su12229767

What are the directions for the future? Why is the subject matter so important and new? What are the research gaps? What's new in this manuscript? (I propose to write in conclusions).

Lines 68, 74 and others ..., I propose to write [29-31], [33-36].

Author Response

Dear Reviewer,

Authors welcome the Reviewer's comment and suggestions for improving the paper. We have made all the changes suggested and hope that the document now fully complies with the Journal's requirements.

CHANGES PROPOSED BY REVIEWER 2:

1.- Reviewer’s comment: Please write in conclusions about the current crisis in tourism caused by the COVID-19 pandemic (including the sharing economy in tourism). Suggested publications:

Roman, M.; NiedzióĹ‚ka, A.; KrasnodÄ™bski, A. Respondents’ Involvement in Tourist Activities at the Time of the COVID-19 Pandemic. Sustainability 2020, 12, 9610. https://doi.org/10.3390/su12229610

Fermani, A.; Sergi, M.R.; Carrieri, A.; Crespi, I.; Picconi, L.; Saggino, A. Sustainable Tourism and Facilities Preferences: The Sustainable Tourist Stay Scale (STSS) Validation. Sustainability 2020, 12, 9767. https://doi.org/10.3390/su12229767

Author’s response: The changes proposed by the Reviewer have been applied to the conclusions of the document, and the corresponding citations have been included in the text (Lines 703-706).

2.- Reviewer’s comment: What are the directions for the future? Why is the subject matter so important and new? What are the research gaps? What's new in this manuscript? (I propose to write in conclusions).

Author’s response: The conclusions have been revised to meet the Reviewer's suggestions (e.g. lines 675-689).

3.- Reviewer’s comment: Lines 68, 74 and others ..., I propose to write [29-31], [33-36].

Author’s response: The changes suggested by the Reviewer have been made.

Reviewer 3 Report

Dear author,

Firstly, his research addresses an emerging business model, the collaborative economy. Predicting consumer behaviour is a critical factor for any company and institution in any sector. Building causal models to predict intention and consumer behaviour is still a great challenge

On the one hand, I found the paper to be overall well written and much of it to be well described. I felt confident that the authors performed careful and thorough field processing. However, I would like to point out some aspects to improve their manuscript. I explain my concerns in more detail below. I ask that the authors specifically address each of my comments in their response.

â… . Major comments

1. One concern I have about the document is with respect to the conceptual delimitation of the Collaborative Economy. This is an emerging phenomenon whose definition is not free of difficulties due to three main reasons: the interest in the study of Collaborative Economics is relatively recent, the literature on this issue is characterized by its fragmentation and interdisciplinarity and a diverse set of terms is frequently used in the literature as synonyms or approximate concepts. All this means that there is no widely accepted definition that can be clarified.

The authors hardly clarify in the text which definition they adopt of the Collaborative Economy, it lacks concreteness. To solve this lack, it is recommended to review the following bibliography:

  • JORGE-VÁZQUEZ, J. La economía colaborativa en la era digital: Fundamentación teórica y alcance económico. In Economía Digital y Colaborativa: Cuestiones Económicas y Jurídicas; Náñez, S.L., Ed.; Università degli Studì Suor Orsola Benincasa. Eurytonpress: Naples, Italy, 2019. ISBN 9788896055915.
  • CHAVES, R. & MONZÓN, J.L. (2018): “La economía social ante los paradigmas económicos emergentes: innovación social, economía colaborativa, economía circular, responsabilidad social empresarial, economía del bien común, empresa social y economía solidaria”, CIRIEC-España, Revista de Economía Pública, Social y Cooperativa, 93, 5-50, DOI: 10.7203/CIRIEC-E.93.12901.

2. The title of the document “Millennials’ Intention to Participate in the Sharing Economy in Tourism: An Integrated Model” is in line with the content of the document. However, I find that the use of the term Millennials to describe the generational approach is not accurate. You indicate in lines 60, 61 and 62 "There is high agreement in the literature that Millennials are people born between 1980 and 2000 that share similar attitudes, perceptions, values, and behaviour [25, 26]". However, when I analyse the sample I find that many of the individuals who participated in the survey were born after 2000. You indicate in line 382 "mean age of the respondents was 20.89 years old". According to the literature these individuals would be part of a new generation named "Generation Z". The following paper is recommended:

  • Dimock, M. (2019). Defining generations: Where Millennials end and Generation Z begins. Pew Research Center17, 1-7.

http://tony-silva.com/eslefl/miscstudent/downloadpagearticles/defgenerations-pew.pdf

To avoid imprecision in the use of the term "Millenials" it is recommended to replace it with "Generation Z" or use an alternative term that is more precise.

â…ˇ.Minor comments:

3. The term eWOM is frequently referred to in the text. It would be helpful if you could indicate its meaning the first time it is used in the text: Electronic Word Of Mouth (eWOM).

4. Bibliographical references should be homogenized. For example in lines 873 or 904 the year does not appear in bold.

In sum, I again thank you for giving me this opportunity to learn from your research project and I wish you the very best. Best Regards,

Author Response

Dear Reviewer,

Authors welcome the Reviewer's comment and suggestions for improving the paper. We have made all the changes suggested and hope that the document now fully complies with the Journal's requirements.

CHANGES PROPOSED BY REVIEWER 3:

1.- Reviewer’s comment: . One concern I have about the document is with respect to the conceptual delimitation of the Collaborative Economy. This is an emerging phenomenon whose definition is not free of difficulties due to three main reasons: the interest in the study of Collaborative Economics is relatively recent, the literature on this issue is characterized by its fragmentation and interdisciplinarity and a diverse set of terms is frequently used in the literature as synonyms or approximate concepts. All this means that there is no widely accepted definition that can be clarified.

The authors hardly clarify in the text which definition they adopt of the Collaborative Economy, it lacks concreteness. To solve this lack, it is recommended to review the following bibliography:

  • JORGE-VÁZQUEZ, J. La economía colaborativa en la era digital: Fundamentación teórica y alcance económico. In Economía Digital y Colaborativa: Cuestiones Económicas y Jurídicas; Náñez, S.L., Ed.; Università degli Studì Suor Orsola Benincasa. Eurytonpress: Naples, Italy, 2019. ISBN 9788896055915.
  • CHAVES, R. & MONZÓN, J.L. (2018): “La economía social ante los paradigmas económicos emergentes: innovación social, economía colaborativa, economía circular, responsabilidad social empresarial, economía del bien común, empresa social y economía solidaria”, CIRIEC-España, Revista de Economía Pública, Social y Cooperativa, 93, 5-50, DOI: 10.7203/CIRIEC-E.93.12901.

Author’s response: The changes suggested by the Reviewer have been applied in paragraph one. One short definition of the sharing economy and the second citation suggested by the Reviewer has been added (Lines 39-42).

2.- Reviewer’s comment: The title of the document “Millennials’ Intention to Participate in the Sharing Economy in Tourism: An Integrated Model” is in line with the content of the document. However, I find that the use of the term Millennials to describe the generational approach is not accurate. You indicate in lines 60, 61 and 62 "There is high agreement in the literature that Millennials are people born between 1980 and 2000 that share similar attitudes, perceptions, values, and behaviour [25, 26]". However, when I analyse the sample I find that many of the individuals who participated in the survey were born after 2000. You indicate in line 382 "mean age of the respondents was 20.89 years old". According to the literature these individuals would be part of a new generation named "Generation Z". The following paper is recommended:

Dimock, M. (2019). Defining generations: Where Millennials end and Generation Z begins. Pew Research Center, 17, 1-7.

http://tony-silva.com/eslefl/miscstudent/downloadpagearticles/defgenerations-pew.pdf

To avoid imprecision in the use of the term "Millenials" it is recommended to replace it with "Generation Z" or use an alternative term that is more precise.

Author’s response:

The authors have considered that the Reviewer's suggestion is correct. Generation Z is indeed very recent and similar to Generation Y (Millennials). However, to be precise the study has been carried out on the incipient Generation Z. Therefore, we have deleted the term "Millennials" from the entire document, including the title, and we have replaced it with "Generation Z", "young consumer" and similar (e.g. Title, Abstract, lines 61-76).

3.- Reviewer’s comment: The term eWOM is frequently referred to in the text. It would be helpful if you could indicate its meaning the first time it is used in the text: Electronic Word Of Mouth (eWOM).

Author’s response:

The changes suggested by the Reviewer have been applied in Lines 210 and 211.

4.- Reviewer’s comment: Bibliographical references should be homogenized. For example in lines 873 or 904 the year does not appear in bold.

Author’s response:

The changes suggested by the Reviewer have been applied, and all references have been reviewed.

Round 2

Reviewer 2 Report

I will accept the manuscript in the current version. Good luck.

Reviewer 3 Report

All suggestions were incorporated. Thanks!